# Personality, Defenses, Mentalization, and Epistemic Trust Related to Pandemic Containment Strategies and the COVID-19 Vaccine: A Sequential Mediation Model

**DOI:** 10.3390/ijerph192114290

**Published:** 2022-11-01

**Authors:** Annalisa Tanzilli, Alice Cibelli, Marianna Liotti, Flavia Fiorentino, Riccardo Williams, Vittorio Lingiardi

**Affiliations:** Department of Dynamic and Clinical Psychology, and Health Studies, Faculty of Medicine and Psychology, Sapienza University of Rome, 00185 Rome, Italy

**Keywords:** COVID-19 pandemic, vaccine hesitancy, personality, defense mechanisms, mentalization, reflective functioning, epistemic trust

## Abstract

Background: The COVID-19 pandemic has considerably influenced all domains of people’s lives worldwide, determining a high increase in overall psychological distress and several clinical conditions. The study attempted to shed light on the relationship between the strategies adopted to manage the pandemic, vaccine hesitancy, and distinct features of personality and mental functioning. Methods: The sample consisted of 367 Italian individuals (68.1% women, 31.9% men; M age = 37, SD = 12.79) who completed an online survey, including an instrument assessing four response styles to the pandemic and lockdown(s), the Personality Inventory for DSM-5-Brief Form, the Defense Mechanisms Rating Scales-Self-Report-30, the Reflective Functioning Questionnaire, and the Epistemic Trust, Mistrust, Credulity Questionnaire. Results: Maladaptive response patterns to pandemic restrictions were related to dysfunctional personality traits, immature defense mechanisms, poor mentalization, and epistemic mistrust or credulity. Moreover, more severe levels of personality pathology were predictive of an extraverted-maladaptive response style to health emergency through the full mediation of low overall defensive functioning, poor certainty of others’ mental states, and high epistemic credulity. Conclusions: Recognizing and understanding dysfunctional psychological pathways associated with individuals’ difficulties in dealing with the pandemic are crucial for developing tailored mental-health interventions and promoting best practices in healthcare services.

## 1. Introduction

The World Health Organization (WHO) recognized the COVID-19 global outbreak as a pandemic in March 2020, raising the health emergency levels due to the increasing infection rates and mortality caused by the high transmissibility of the acute respiratory syndrome coronavirus 2 [1]. As a result, governments in different countries have been forced to introduce severe restrictive measures to prevent the spread of SARS-CoV-2 [2]. Over two years, the pandemic and its related containment strategies profoundly affected people’s lives worldwide, causing financial instability, uncertainty about the future, and negative consequences for psychological well-being [3,4,5]. Evidence showed that the COVID-19 threats and the mitigation measures that were adopted were associated with greater levels of worry, self-isolation, loneliness, frustration, anger, and fear [6,7], as well as post-traumatic stress disorder (PTSD) symptoms [8], and suicidal ideation [9,10]. 

The detrimental psychological effects of this socio-health emergency have also represented a relevant risk factor for developing and exacerbating several mental conditions [6,11,12,13,14]. Reviews and meta-analyses [15,16] highlighted an alarming increase in depressive and anxiety disorders, as well as in overall psychological distress, among the general population worldwide (e.g., Australia [17], China [18,19], England [20]; France [21], Italy [22,23]; Turkey [24], the United States [25], etc.).

By the end of 2020, the COVID-19 vaccine—developed to provide immunity against SARS-CoV-2—had begun to be administered. Many countries immediately implemented effective distribution plans, prioritizing the vaccination of healthcare workers and individuals at high risk of exposure and transmission or severe complications after contracting the infection. In Italy, in approximately April 2021, the vaccination campaign was extended to the general population. The Italian governmental bodies and other organizations tried to adopt many strategies to incentivize vaccination; however, protests, demonstrations, and strikes rapidly spread, similarly to what happened in other countries. Some anti-vaccine activists, seemingly influenced by a conspiracy mindset, have also developed unfounded beliefs about COVID-19 vaccines based on misunderstood or misrepresented (un)scientific or religious theories.

The present research aims to obtain a better understanding of the complex psychological processes that may have influenced the Italian people’s behavioral responses to the COVID-19 pandemic through a psychodynamically oriented perspective (cf., [26,27]). In detail, this study has sought to extend the knowledge from previous empirical investigations about the impact that some distinct characteristics of personality, as well as defense mechanisms and other mental capacities (such as mentalization and epistemic trust), exert on individuals’ adaptation to the health emergency created by the SARS-CoV-2. We were interested in investigating how these domains of psychological functioning might have affected the individuals’ response to the containment measures and vaccine hesitancy during the second wave of COVID-19 (when the effects of the vaccine were less known).

### 1.1. The Role of Personality and Mental Functioning Capacities in Coping with the Pandemic

Recent studies showed that personality traits—considered in terms of relatively stable individual characteristics that may predict or explain specific patterns of behaviors or conducts in dealing with life experiences—played a relevant role in managing the healthcare emergency and enforced restrictions due to the pandemic [28,29,30]. 

Research, primarily based on the Big Five personality framework [31] or the HEXACO model [32,33], revealed that some dimensions strongly influence individual psychological and behavioral reactions to COVID-19 [34]. People with high levels of neuroticism present a more pessimistic vision in response to COVID due to their tendency to experience the impact of stressful events with intense concerns and distress. They are also more prone to developing anxiety, depression, and post-traumatic stress symptoms [30,35,36,37,38,39]. Conversely, individuals with high degrees of extraversion, conscientiousness, and agreeableness tend to manifest greater optimism and activate adaptive strategies to preserve their psychological well-being. They are also more likely to respect social distancing and all the precautions and governmental measures against COVID-19, minimizing infection risk [39,40,41,42,43]. Although extraversion is generally related to lower distress and better adaptive behaviors to the pandemic, studies have demonstrated that the restrictions imposed during the COVID-19 lockdown may create difficulties in highly extraverted individuals because of the forced reduction in social interactions they imposed [38,44]. Evidence also revealed that the impact of openness to new experiences is controversial, probably due to the lack of a widely shared, conceptually consistent, and clinically sensitive definition of this personality dimension [45,46]. While some studies indicated that this personality trait is associated with more compliance with containment measures for COVID-19 [41], others pointed out that individuals with higher openness levels tend to have a lower perception of risk, thus being more prone to act recklessly [47].

According to a psychodynamically oriented perspective, other relevant factors involved in managing the psychological impact of stressful, threatening, or traumatic events are represented by defense mechanisms—defined as mostly unconscious mental processes that protect the individual against distress and mediate the emotional reactions to conflicts generated by the clash between needs, impulses, desires, and affects on the one hand, and internal prohibitions and/or conditions of external reality on the other ([48,49,50], cf., [51]). Mental health is closely related to the ability to use a variety of adaptive defenses in challenging contexts; an excessive resort to immature defenses is a risk factor for the development of several psychopathological conditions [52,53,54], personality syndromes in particular [55,56,57,58,59].

Some studies examining psychological difficulties related to the COVID-19 pandemic proved that adaptive defenses—such as altruism, sublimation, and humor—mitigate anxiety over the viral outbreak and promote resilience and adaptation without distortions of reality [60,61]. Conversely, other empirical investigations demonstrated that fears, anxieties, and concerns from preventive measures related to the COVID-19 pandemic elicit the activation of maladaptive defense mechanisms [34,60,62,63,64]. The use of immature defenses was also predictive of psychological distress and psychiatric conditions during COVID-19 [65]; lower overall defensive functioning was associated with higher depressive and PTSD symptoms [63].

Mentalization is another meaningful dimension in determining individuals’ strategies to adjust to reality. It is a specific mental activity related to the perception and interpretation of human behavior in terms of mental states motivated by intentions—e.g., needs, desires, feelings, beliefs, and goals [66,67]. The ability to mentalize enables individuals to interact with others, make accurate sense of their own and others’ mental states, and distinguish internal and external realities without distortions [68]. When distressed and exposed to unsafe or threatening situations, individuals may regress to prementalizing modes and lose their ability to understand the perspective of others. Some authors supposed a collapse in mentalizing due to the severe mental health consequences of the COVID-19 pandemic and a subsequently increased vulnerability to mental suffering [69], suggesting the need to examine this dimension carefully. Other research pointed out that mentalizing operates as a protective function by promoting resilience to the impact of stress and threats [70]. However, to date, the role of mentalization in influencing individuals’ adaptation to the pandemic has scarcely been investigated. 

The dimension of epistemic trust (ET)—strongly related to mentalization [27]—refers to the capacity of an individual to consider the knowledge conveyed by others as significant, relevant to the self, and generalizable to other contexts [71]. The latest conceptualizations emphasize the role that disruptions in ET play in undermining resilience and increasing the risk of developing mental health problems [27,72,73]. Such disruptions may manifest through two different maladaptive epistemic stances: epistemic mistrust (EM, i.e., a rigid and pervasive hypervigilant stance toward new information coming from others) and epistemic credulity (EC, i.e., an indiscriminate trust in others, making the individual vulnerable to manipulation or mistreatment). EM and EC may represent significant factors influencing the individuals’ psycho(patho)logical and behavioral responses to stressful or traumatic situations such as the pandemic, given that they may hinder the ability to acquire and accommodate new information in a way that supports resilient functioning and adaptation [74]. Focusing on these dimensions could be particularly valuable for understanding opposition to vaccination (a phenomenon already described as one of the top ten global threats by the WHO in 2019; [75]) and the more general tendency to rely upon conspiracy theories and misinformation, with consequences such as the unwillingness to act in accordance with safety measures against COVID-19. Research has found that the conspiracy mentality is associated with a tendency to manifest mistrust towards others independently from untrustworthiness cues [76] and that people with a conspiratorial mindset are likely to “turn the trust-the-expert heuristic upside down” [77], cf. [42]). Nevertheless, to date, only one study has explored the potential role of ET in this global healthcare and social emergency, suggesting that EM and EC are associated with pandemic conspiracy theories [78].

### 1.2. The Current Study: Aims and Hypotheses

Italy was the first European country to deal with the COVID-19 emergency, adopting immediate protective strategies to counter the virus’s circulation. The first lockdown was ordered over the national territory on 9 March 2020. During the pandemic, the need to contain the new waves of contagion and prevent the spread of many local outbreaks at higher risk (called “red areas”) led the government to provide for other total or partial lockdowns, as well as a series of necessary precautions and safeguards to ensure the safety of citizens. 

The present study was conducted during the second wave of contagion (between March and April 2021) to investigate, from a psychodynamic perspective, the subjective experience of Italian citizens facing the COVID-19 pandemic and all the governments’ issued containment measures. For this purpose, we created four prototypical descriptions of behavioral styles triggered by lockdown(s) and other restrictions. These patterns were derived by intersecting orthogonal lines that represent two crucial dimensions of psychological and personality functioning: introversion/extroversion and adaptation/maladaptation to reality. This grid is partially inspired by Kernberg’s [79] personality dimensional model, in which the X-axis represents the well-established dimension, i.e., the continuum of introversion/extraversion. In our model, the Y-axis represents the polarities of adaptation/maladaptation, differentiated based on the defense mechanisms that were adopted [50] and reality-testing abilities [79]. 

As depicted in Figure 1, four response styles to the pandemic and lockdown(s) (named RSPL) were identified: “A” or introverted/maladaptive style, “B” or introverted/adaptive style, “C” or extraverted-adaptive style, and “D” or extraverted-maladaptive style. Overall, these clinical vignettes (see Appendix A) describe different ways of feeling, thinking, and behaving that are activated to face the distress related to the pandemic (and its restrictive measures) and withstand the strong emotional impact of this collective traumatic experience.

As mentioned above, the present research aims to extend previous empirical investigations and fill some gaps in the scientific literature by exploring the associations between specific ways of dealing with the pandemic (from an affective, cognitive, and behavioral point of view) and personality characteristics, defensive functioning, mentalization, and epistemic trust within the Italian context. We followed a psychodynamically oriented approach––based on a strong interconnection of all the aforementioned psychological dimensions (for a more thorough discussion, cf. [26,27])––focusing on four main goals:Examine the relationship between response styles to pandemic containment measures and vaccine hesitancy. Consistent with the literature (e.g., [80]), we expected that introverted and extraverted maladaptive styles (A and D) would be related to COVID vaccine hesitancy.Investigate the associations between different response patterns to the COVID emergency and individual personality traits (evaluated according to the Alternative Model of Personality Disorders; [51]). Consistent with the previous research mentioned above (e.g., [30,35,81,82]), we hypothesized that higher levels of detachment would be significantly associated with an introverted-maladaptive style. In contrast, we expected greater negative affectivity, antagonism, and disinhibition levels to be significantly associated with an extraverted-maladaptive style.Explore the relationship between distinct reactions to the pandemic (and its containment measures) and defensive functioning, identifying whether specific defense mechanisms were related to specific styles of adaptation. In line with empirical studies in the literature [36,60], we hypothesized that lower overall defensive functioning and more immature defense levels would be significantly associated with more maladaptive and dysfunctional response patterns to the social-health emergency (i.e., A and D styles). In more detail, despite the few studies in this research field [61], we expected that: (a) introverted-maladaptive style would be related to defense mechanisms such as an isolation of affects, dissociation, projection, and autistic fantasy; (b) introverted-adaptive style would be related to sublimation and self-observation; (c) extraverted-adaptive style would be related to humor, altruism, and affiliation; (d) extraverted-maladaptive style would be related to omnipotence, denial, splitting of self-image and others’ image, and acting out.Examine the associations among response styles to the pandemic, mentalization, and all the dimensions of ET (including EM and EC). Trying to bridge a gap in the empirical literature [69,78], we hypothesized that lower levels of mentalization and high degrees of mistrust and credulity would be significantly related to the most maladaptive patterns of responding to emergency difficulties.Investigate, in an exploratory analysis, the relationships (in terms of direct and indirect effects) of all the psychological dimensions included in the study on individual dysfunctional reactions to the pandemic. In line with previous research, we hypothesized that the global level of personality pathology would be related to more maladaptive patterns (A and D styles) when dealing with a socio-health emergency and its traumatic consequences through the partial mediation of poor defensive functioning, low ability to mentalize, and high EM.

## 2. Materials and Methods

### 2.1. Procedures

Participants were recruited using an online survey (hosted on SurveyMonkey) that was launched on 30 March 2021 and remained open until 30 April 2021. Individuals were approached via advertisements posted on websites and social networks (e.g., Facebook, Instagram, Twitter) and via email. According to the inclusion/exclusion criteria, participants had to: (1) provide consent to data-processing for research purposes; (2) be at least 18 years old; (3) live or have lived in Italy during the entire period of the pandemic. Participation was voluntary and completely anonymous to guarantee privacy. Before participating, all the individuals provided informed consent electronically. The research protocol was approved by the Ethical Committee of the [Department of Dynamic and Clinical Psychology, and Health Studies, Faculty of Medicine and Psychology, Sapienza University of Rome], Prot. n. 0000548/2021. 

### 2.2. Measures

*Clinical Questionnaire*: A self-report questionnaire was constructed to obtain general information about the participants (including their educational level or socioeconomic status), their COVID-19 experience (for example, possible changes in their employment position due to the pandemic or the bereavement of friends or family members sick with SARS-CoV-2), and attitudes toward the Italian vaccination program (such as degree of agreement with vaccination policies and levels of trust/confidence in the positive effects of the vaccine). A global index on attitudes toward COVID-19 vaccines was derived from these specific questions.

*Response Styles to the Pandemic and Lockdown(s)*: The RSPL, an ad-hoc instrument developed by our group [83], consists of four vignettes representing distinct affective, cognitive, and behavioral styles of response to the pandemic and the restrictive measures imposed during the lockdown(s). The response patterns were derived from the intersection of two psychological dimensions: introversion vs. extraversion and adaptation vs. maladaptation to reality. These four patterns describe individuals presenting with distinct intra- and interpersonal characteristics, as follows: *introverted-maladaptive style* (A); *introverted-adaptive style* (B); *extraverted-adaptive style* (C); and *extraverted-maladaptive style* (D) (see Appendix A). For example, maladaptively extraverted individuals (D style) tend to show a reckless disregard for the rules and restrictions imposed by the government to mitigate the pandemic effects. They seem to be indifferent to the rights and safety of others. They are impulsive and appear indifferent to the consequences of their actions. These individuals seem unable or unwilling to modify their behaviors, not respecting any limits and failing to adapt to reality in an appropriate and socially acceptable way. 

According to a categorical approach, all the participants were asked to identify their way of reacting to the stressful situation related to the COVID-19 contagion and its containment strategies. Then, following a dimensional approach, participants had to score on a 7-point Likert scale, ranging from 1 (*not alike at all*) to 7 (*very much alike*), how each description matched their subjective and interpersonal experience of the socio-health emergency.

*Personality Inventory for DSM-5–Brief Form–Adult*: The PID-5-BF [84] is a 25-item self-report instrument assessing personality traits was developed to operationalize the five personality trait domains (including negative affectivity, detachment, antagonism, disinhibition, and psychoticism) according to the Alternative Model of Personality Disorders (AMPD) included in Section III of the DSM-5 [51]. The respondents must assess each item on a 4-point Likert scale ranging from 0 (*very false or often false*) to 3 (*very true or often true*). Higher scores in each domain indicate greater dysfunction in the corresponding personality trait. The overall level of personality dysfunction is measured by dividing the raw score by the total number of items in the measure. The instrument shows good levels of validity and reliability [85]. In the study, the PID-5-BF scales demonstrated good internal consistency [86]: negative affectivity (α = 0.70), detachment (α = 0.79), antagonism (α = 0.70); disinhibition (α = 0.68), and psychoticism (α = 0.80).

*Defense Mechanisms Rating Scales-Self-Report-30*: The DMRS-SF-30 [87] is a 30-item self-report questionnaire describing the hierarchy of defense mechanisms developed by Perry [50]. The pool of 30 items was extracted from the observer-rated Defense Mechanisms Rating Scales Q-sort version [88]. The respondents must assess each item on a 5-point Likert scale, ranging from 0 (*not at all*) to 4 (*very often/much*). The DMRS-SR-30 provides distinct quantitative scores related to the overall defensive functioning (ODF), seven defense levels, and individual defenses. The hierarchically ordered defense levels ranged from the most to the least adaptive and included: (a) *high-adaptive or mature* (including affiliation, altruism, anticipation, humor, self-assertion, self-observation, sublimation, and suppression), (b) *obsessive* (including undoing, intellectualization, and isolation of affects), (c) *neurotic* (including repression, dissociation, reaction formation, and displacement), (d) *major image-distorting* (including idealization of self and others’ images, devaluation of self and others’ images, and omnipotence), (e) *disavowal* (including denial, rationalization, projection, and autistic fantasy), (f) *major image-distorting* (including splitting of self and others’ images, and projective identification), and (g) *action defenses* (including acting out, passive aggression, and help-rejecting complaining). Preliminary analyses of the psychometric properties of the DMRS-SR-30 provided good results in terms of internal consistency (e.g., Cronbach’s alpha for ODF = 0.89), criterion, concurrent, convergent, and discriminant validity [89].

*Reflective Functioning Questionnaire*: The RFQ [90] is an 8-item self-report questionnaire developed to assess mentalizing abilities, i.e., the individuals’ capacity to understand their own and others’ reactions and behaviors in terms of underlying mental states (e.g., feelings, desires, wishes, goals, and attitudes). The respondents must assess each item on a 7-point Likert scale, ranging from 1 (*completely disagree*) to 7 (*completely agree*). This measure evaluates two dimensions: *certainty* (RFQ_C) about one’s and others’ mental states (e.g., “I always know what I feel”) and *uncertainty* (RFQ_U) about them (e.g., “People’s thoughts are a mystery to me”). The Italian validation of the RFQ demonstrates high levels of validity and reliability [91]. In the study, the RFQ scales showed good internal consistency [85]: RFQ_C (α = 0.74) and RFQ_U (α = 0.71). 

*Epistemic Trust, Mistrust, Credulity Questionnaire*: The ETMCQ [71] is a 15-item self-report questionnaire assessing the capacity of the individual to consider information conveyed by others as significant, relevant to the self, and generalizable to other contexts. The respondents must assess each item on a 7-point Likert scale, ranging from 1 (*strongly disagree*) to 7 (*strongly agree*). This instrument showed a three-factor structure consisting of distinct dimensions: (a) *trust*, referring to an adaptive stance in relatively benign social circumstances, in which the individual is appropriately open to opportunities for social learning (e.g., “I find information easier to trust and absorb when it comes from someone who knows me well”); (b) *mistrust*, reflecting a tendency to treat any source of information as unreliable or ill-intentioned, and reject or avoid any influence of communication from others (e.g., “If you put too much faith in what people tell you, you are likely to get hurt”); and (c) *credulity*, referring to a pervasive lack of discrimination and clarity about one’s position that promotes a vulnerability to misinformation and potential risk of exploitation (e.g., “When I speak to different people, I find myself easily persuaded even if it is not what I believed before”) [71]. In the study, the ETMCQ scales showed good internal consistency [86]: ET (α = 0.70), EM (α = 0.70), and EC (α = 0.76). 

### 2.3. Statistical Analysis

Statistical analyses were performed using SPSS 26 for Windows (IBM, Armonk, NY, USA). The relationships between response styles to pandemic containment measures (assessed using the dimensional scores obtained from RSPL vignettes) and socio-demographic variables (e.g., gender, age, geographic provenance) were analyzed using Multivariate Analysis of Variance (MANOVA) and bivariate correlations (r, two-tailed). To explore associations between responses to COVID-19 emergency, vaccine hesitancy, and individual psychological dimensions considered in the study, bivariate correlations were also carried out by considering an overall index of pro-vaccine attitudes, personality traits and global functioning (assessed with PID-5-BF), defense mechanisms (measured with the DMRS-SR-30), scales of reflective functioning (evaluated using the RFQ), and ET dimensions (assessed with the ETMCQ).

A sequential multiple mediational model was performed to assess the effect of global personality functioning on the most maladaptive patterns of responses to the pandemic (A and D styles) through the mediation role of overall defensive functioning (ODF), certainty and uncertainty about mental states (RFQ_C and RFQ_U), and ET, EM, and EC. The tested mediation model was estimated using the PROCESS macro for SPSS [92], according to ordinary least-squares (OLS) regression (Model 6). Following Taylor et al. [93], before performing the model, a series of multiple regression analyses were estimated to verify whether the criteria of the sequential mediation model were satisfied. The significance of the model was assessed using the bootstrapping method (CI = 95%), which shows a high power for hypothesis testing and does not assume a normal sample distribution [94]. Bootstraps were set at 5000, and effects were considered significant when the CI excluded zero.

## 3. Results

### 3.1. Participant Characteristics

The sample consisted of 367 Italian participants, of which 250 (68.1%) were women and 117 (31.9%) were men; the average age was 37 years (SD = 12.79, range = 18–70). Their provenance was northern Italy (56.7%), central Italy (36.2%), and southern Italy and islands (7.1%). The instruction levels ranged from upper secondary education (22%) to higher educational levels (including university and post-graduate degrees) (45%). Most participants were single (36.8%) or married (24.8%). The socioeconomic status was middle (79.3%). One-hundred-and-sixty-four participants (44.69%) had public employment, 67 (18.27%) were freelancers, and 104 (28.34%) were students. Other characteristics of the sample are shown in Table 1.

Regarding the categorical classification of response patterns to COVID containment strategies (see RSPL), 30 participants (8.2%) recognized themselves in A style (introverted/maladaptive), 170 participants (46.3%) in B style (introverted/adaptive), 142 participants (38.7%) in C style (extraverted/adaptive), and 25 participants (6.8%) in D style (extraverted/maladaptive).

Analyses that examined the relationships between response styles to pandemic containment measures (assessed using RSPL dimensional scores) and specific socio-demographic characteristics (including gender, geographic provenance, and socioeconomic level) did not find significant results. MANOVA showed no effect of gender on individuals’ reactions to COVID-19 emergency, Wilks’s λ = 0.99, *F*(4, 362) = 1.189, *p* = 0.32, η^2^ = 0.013, Cohen’s *d* = 0.23. In more detail, female participants (*M_introverted/maladaptive style_* = 2.92, *SD* = 1.50; *M_introverted/adaptive style_* = 4.08, *SD* = 1.34; *M_extraverted/adaptive style_* = 4.04, *SD* = 1.37; *M_extraverted/maladaptive style_* = 2.05, *SD* = 1.43) did not report significantly different response patterns to male participants (*M_introverted/maladaptive style_* = 3.00, *SD* = 1.57; *M_introverted/adaptive style_* = 4.28, *SD* = 1.20; *M_extraverted/adaptive style_* = 3.79, *SD* = 1.42; *M_extraverted/maladaptive style_* = 1.86, *SD* = 1.30). 

Moreover, analysis showed no impact of geographic provenance on response styles to the pandemic, Wilks’s λ = 0.97, *F*(8, 722) = 1.390, *p* = 0.19, η^2^ = 0.015, Cohen’s *d* = 0.25. Participants from northern Italy (*M_introverted/maladaptive style_* = 2.83, *SD* = 1.52; *M_introverted/adaptive style_* = 4.19, *SD* = 1.26; *M_extraverted/adaptive style_* = 3.88, *SD* = 1.40; *M_extraverted/maladaptive style_* = 2.07, *SD* = 1.49) did not report significantly different responses to those from central Italy (*M_introverted/maladaptive style_* = 3.08, *SD* = 1.54; *M_introverted/adaptive style_* = 4.11, *SD* = 1.35; *M_extraverted/adaptive style_* = 4.14, *SD* = 1.34; *M _extraverted/maladaptive style_* = 1.90, *SD* = 1.22) or from southern Italy and the islands (*M_introverted/maladaptive style_* = 3.12, *SD* = 1.42; *M_introverted/adaptive style_* = 4.00, *SD* = 1.39; *M_extraverted/adaptive style_* = 3.65, *SD* = 1.47; *M_extraverted/maladaptive style_* = 1.85, *SD* = 1.35). 

No effect of socio-economic level was also revealed; Wilks’s λ = 0.97, *F*(8, 722) = 1.385, *p* = 0.20, η^2^ = 0.015, Cohen’s *d* = 0.25. Participants from the upper class (*M _introverted/maladaptive style_* = 2.88, *SD* = 1.39; *M_introverted/adaptive style_* = 4.38, *SD* = 1.16; *M_extraverted/adaptive style_* = 4.18, *SD* = 1.27; *M _extraverted/maladaptive style_* = 1.56, *SD* = 1.02) did not report significantly different responses than those from middle (*M_introverted/maladaptive style_* = 2.91, *SD* = 1.51; *M_introverted/adaptive style_* = 4.13, *SD* = 1.31; *M_extraverted/adaptive style_* = 3.95, *SD* = 1.39; *M_extraverted/maladaptive style_* = 2.01, *SD* = 1.40) and lower classes (*M_introverted/maladaptive style_* = 3.24, *SD* = 1.72; *M_introverted/adaptive style_* = 4.05, *SD* = 1.32; *M_extraverted/adaptive style_* = 3.86, *SD* = 1.51; *M_extraverted/maladaptive style_* = 2.21, *SD* = 1.54). Finally, no significant relationships were found between age and pandemic emergency responses, except for the negative correlation with extrovert–adaptive style (*r* = −0.24, *p* < 0.001).

### 3.2. Relationship between Response Styles to the COVID-19 Pandemic and Vaccine Hesitancy

The first aim of the present study was to explore the associations among the four response patterns to the pandemic (using the dimensional approach of RSPL) and the attitudes toward vaccination policies. Partially confirming our hypotheses, bivariate correlations showed that a global pro-vaccine index was negatively associated with extraverted-maladaptive style (*r* = −0.42, *p* ≤ 0.001), and positively associated with introverted-maladaptive (*r* = 0.12, *p* ≤ 0.05), introverted-adaptive (*r* = 0.21, *p* ≤ 0.001), and extraverted-adaptive styles (*r* = 0.14, *p* ≤ 0.01).

### 3.3. Relationship between Response Styles to the COVID-19 Pandemic and Personality 

The second aim of the study was to examine the relationship between response styles to COVID-19 and individual personality traits (Table 2). 

The results were partially consistent with our expectations. The introverted-maladaptive style was positively associated with negative affectivity and detachment personality traits, whereas the extraverted-maladaptive style was related to disinhibition and antagonism traits; a trend toward significance was found between this latter style and psychoticism traits. Finally, a significant correlation between the extraverted-maladaptive style and global index of personality pathology was found. 

### 3.4. Relationship between Response Styles to the COVID-19 Pandemic and Defensive Functioning

The third aim of the study was to investigate the associations between different styles of dealing with the social-health emergency and defensive functioning (Table 3). 

In line with our hypotheses, the results showed that the extraverted-maladaptive style was associated with lower ODF. In addition, mature defense levels correlated negatively with the most dysfunctional response patterns (introverted- and extraverted-maladaptive styles, A and D) and were instead positively associated with the most adaptive ones (introverted- and extraverted-adaptive styles, B and C). Moreover, the extraverted-maladaptive style correlated positively with action, major-, and minor-image distorting defense levels (in particular, acting out, projective identification, and omnipotence), whereas introverted-maladaptive style was related to disavowal defense level (especially autistic fantasy and projection) and dissociation.

### 3.5. Relationship between Response Styles to COVID-19 Pandemic, Mentalization, and Epistemic Trust 

The fourth aim of the study was to explore the relationship between distinct styles of coping with the pandemic and its mitigation measures, mentalization levels, and epistemic trust (Table 4). The findings showed that introverted- and extraverted-maladaptive patterns were negatively related to certainty about mental states. Moreover, the extraverted-adaptive style was associated with ET; conversely, introverted- and extraverted-maladaptive styles were positively associated with EM and EC, respectively.

### 3.6. Maladaptive Response Styles to the COVID-19 Pandemic and Its Related Containment Strategies: A Sequential Mediation Model

The last aim of the study was to explore the relationships among all the psychological characteristics considered and examine the direct and indirect effects of global personality pathology on the more maladaptive reactions to the pandemic (i.e., A and D styles). A series of multiple regression analyses were estimated to verify whether the criteria of the sequential mediation model were satisfied. Only the mediation sequential model considering style D (extraverted-maladaptive) as a dependent variable met the criteria. As depicted in Figure 2, all the indirect effects among the key variables of the model were statistically significant; thus, the level of global personality pathology predicted an extraverted-maladaptive response style to the pandemic through the full mediation of low defensive functioning, impaired mentalization, and high EC. 

## 4. Discussion

The COVID-19 pandemic has considerably influenced all domains of people’s life (e.g., physical, psychological, economic, political, and social). Evidence in the mental health field has documented a high increase in psychiatric disorders and overall psychological distress [12,14,15,16]. These data reflect the difficulties individuals have in dealing with the global emergency and the consequences of the enforced implementation of social-health policies promoted by governments to contain the spread of SARS-CoV-2 (e.g., [3,11]). The present study attempted to shed light on the relationship between the specific strategies adopted to manage the pandemic and distinct aspects of individual mental functioning. Our aim was to provide a comprehensive view of the psychological processes involved in the adjustment to unexpected, constantly changing, and traumatic circumstances related to the global emergency. 

Overall, the findings showed that distinct patterns of affective, cognitive, relational, and behavioral responses to the pandemic (moving along two dimensions: adaptation vs. maladaptation and introversion vs. extraversion; see Appendix A) were significantly associated with specific attitudes toward COVID vaccination, personality features (Table 2) and mental-functioning capacities, especially in terms of defense mechanisms (Table 3), mentalization, and ET (Table 4). Moreover, an accurate investigation of the relationship among these psychological characteristics allowed for us to identify their impact on the most maladaptive response to the health emergency (Figure 2). Recognizing and understanding these dysfunctional psychological pathways represent crucial factors in developing personalized mental-health interventions and promoting best practices in healthcare services [95].

Notably, partially confirming our first hypotheses, the study found that only the extraverted-maladaptive style (D) is related to vaccine hesitancy. Consistent with previous research [80], these individuals tend to show a lack of trust in COVID-19 vaccine safety, manifesting skepticism about its efficacy. Secondly, this research aimed to explore whether there were significant associations between personality traits and individual response patterns characterized by more difficulties in coping with anxiety, stress, and the sense of threat linked to the pandemic and its related containment measures. In line with the empirical literature (e.g., [3,39,96,97]), the results showed that introverted- and extraverted-maladaptive styles (A and D) presented more pathological personality features. Partially supporting our hypotheses, the introverted-maladaptive style (A) was associated with high levels of negative affectivity and detachment. In other words, individuals more strongly characterized by anxiety and emotional lability, along with withdrawal and avoidance of intimacy, showed diminished mutuality within close relationships during the pandemic due to fear of being engaged in activities involving interpersonal contact; moreover, they manifested more rigid attitudes, strictly adhering to the government-imposed measures [35,82]. These findings are consistent with previous studies revealing how high levels of neuroticism and introversion contributed strongly to worse pandemic management; moreover, these personality traits were found to predict more severe degrees of loneliness, anxiety, and depression during the COVID-19 emergency [98,99].

Conversely, the extraverted-maladaptive style (D) was related to antagonism, disinhibition, and high levels of personality pathology. These findings seem to support other studies pointing out how lower agreeableness and conscientiousness hinder the commitment to pandemic containment measures [36,40,82,100,101] and promote a reduced adherence to prevention guidelines related to COVID-19 [41]. Individuals who show disregard for the needs of others, who tend to be grandiose and impulsive, and to adopt antisocial behaviors, present lower compliance with protective behaviors enforced to contain the COVID-19 contagion [35,81]. It is likely that the incapacity of elaborating and containing frustration might have led them to be unable to tolerate the restrictions during lockdown(s) and to get hostile, seeking risk irresponsibly, and thus showing substantial unreliability and breaching the rules imposed by the pandemic. In addition, the trend toward significance with respect to the association between style D and psychoticism traits could reflect these individuals’ lack of connection to their life context. As a result, they may face considerable difficulties in preserving their reality testing, showing a greater tendency to not respect the government’s recommended health measures [102,103].

The third objective of our study was to investigate the associations between individual response styles to COVID-19 containment measures and defense mechanisms. The extraverted-maladaptive style (D) was strongly associated with lower overall defensive functioning and immature defenses (i.e., action and both major- and minor-image distortion levels), while the introverted-maladaptive style (A) was related to disavowal defense level (see the hierarchy of defense mechanisms; [48]). These findings seem to support previous studies highlighting that excessive reliance on dysfunctional defense mechanisms is linked to a worse ability to effectively cope with stressful and emotionally charged events, as well as to greater psychological distress and, potentially, the development of clinical conditions [36,60,61]. It is well-known that immature defenses (belonging to action and both major- and minor-image distortion levels) predict the exacerbation of episodes of depression, self-destructive conducts, and suicidal attempts or ideation [48,104]. Similarly, action and minor image-distorting defenses are related to substance abuse and antisocial behaviors [105,106].

Looking at the findings depicted in Table 3 in more detail, it is important to note the strong connection between the extraverted-maladaptive style (D) and defenses such as acting-out, projective identification, and omnipotence. When faced with stressful experiences and painful conditions, these individuals seem to feel the need to radically or partially distort reality to protect themselves from feelings of weakness, fragility, and powerlessness. Thus, they may adopt potentially dangerous behaviors and violate shared rules and regulations, regardless of the consequences to one’s and others’ health [60]. However, the introverted-maladaptive style (A) showed a strong association with autistic fantasy, projection, and dissociation. It is likely that maladaptively introverted individuals deal with traumatic or stressful events related to the pandemic by refusing to acknowledge aspects of reality and their experiences that are too frightening and threatening [61]. Their tendency to retire from reality through autistic fantasy can encourage an urge to find shelter in an unrealistic yet more gratifying dimension, withdrawing from the feelings of helplessness and frustration caused by the pandemic and its restrictions. Their isolation and retreat can also be seen as a defense against an externally projected sense of danger (cf. [48]), whilst the use of dissociation could reflect their inability to face the overwhelming experience of such unexpected and adverse events as the ones created by the pandemic (e.g., [60,63]), especially during the first COVID waves. 

Consistent with our hypotheses, the most adaptive response patterns to the pandemic and its restrictions (B and C styles) were associated with highly adaptive defenses (Table 3). Even in the face of high-arousal situations, these defenses enable individuals to maintain a heightened awareness of their internal experience, activating evolutionary resources and resilience [36,60]. When confronted with severe restrictions due to the pandemic, adaptively introverted individuals seem to use sublimation by channeling their potentially negative feelings and impulses into creative activity or socially acceptable behaviors, whereas adaptively extraverted people resort to affiliation and humor. They may manifest their emotional attachment needs by seeking support from others and trying to relieve their tension and frustration by emphasizing the ironic aspects of pandemic-related experiences (cf. [48]).

The fourth aim of the study was to investigate whether pandemic-triggered response patterns were related to severe impairments of mentalization and ET. Consistent with recent studies showing the interplay among mentalization, defenses, and emotional regulation [107], we identified a negative association between both introverted- and extraverted-maladaptive styles (A and D) and certainty of others’ mental states (RFQ_C). Recent contributions (e.g., [69]; see also [70]) pointed out that the pandemic has weakened the capacity of individuals to infer, understand, and interpret the actions of others, to symbolize meaningful experience affectively (rather than somatically or behaviorally), and to effectively use such experience for both self-regulation and interpersonal interactions. The negative correlations between maladaptive styles of response to the pandemic and the RFQ_C scale––combined with their correlations with immature defense mechanisms––seem to suggest that individuals with A and D styles are characterized by “pretend mode”, defined as a mental state in which one attempts to avoid confrontation with their internal or external reality [108,109]. Pretend mode often accompanies a form of hypomentalizing associated with pseudomentalization, in which the individuals formulate apparently elaborate narratives that actually represent rigid and repetitive interpretations about themselves’ and others’ behaviors and mental states, frequently having a self-serving function and either detached from affective experiences or affectively overwhelmed [110]. These individuals may perceive others’ attempts to promote a mentalizing attitude as somehow intrusive or hostile; therefore, they may recur to immature defense mechanisms (e.g., devaluation or acting out) for rebuffing such attempts, especially when they also present disruptions in their ability to trust information coming from others. Indeed, the socio-health emergency caused by COVID-19 fostered a spread of prementalizing acting-outs (including domestic violence and emotional abuse;, e.g., [111]), which strongly correlated with lower levels of resilience and an overall worsening in the ability to cope with psychological distress [36,60]. Although needing further investigation, our findings appear crucial to the development effective interventions, not only within a therapeutic context but also in the production of guidelines for behavior during pandemics or other adverse situations and, more broadly, in social communication.

Moreover, we found a significant association between response patterns to COVID-19 restrictions and all the ET dimensions, closely connected to mentalization ability and emotion regulation [112,113]. Adaptively extraverted individuals presented a greater capacity to rely on information selected in a significant and sufficiently reliable interpersonal context, showing good levels of adaptation. Conversely, maladaptively introverted individuals manifested high levels of EM, tending to treat any source of information as unreliable and ill-intentioned and to reject any influence by communications from others [71]. Considering their typical use of projection (see Table 3), this tendency seems to reflect the propensity to falsely attribute negative contents to socially acquired knowledge [78]. Similarly, maladaptively extraverted individuals were characterized by greater levels of EC, reflecting their incapacity to discern the (un)reliability of the sources of knowledge, crediting sensationalistic and non-scientifically supported information ([71], cf. [114]). In addition, these people––who tend to believe conspiracy theories about COVID-19––showed greater skepticism toward official reports and were less willing to receive the vaccine or believe in the safety of vaccination practice [78]. These individuals also presented a lesser capacity for mentalizing. Therefore, they seem less able to reflect on complex explanatory models of behaviors and mental states, and more inclined to produce simplistic explanations. When associated with other psychopathological markers (such as those found in our study regarding defense mechanisms), hypomentalizing also seems to be related to an unstable self-image and sense of identity [115]. Individuals with low reflective functioning abilities may, therefore, tend to believe in theories and explanations that can offer them a stronger sense of personal identity and emotional stability, regardless of the reliability of such information. This is consistent with studies finding that high scores on the RFQ_C scale are related to higher abilities to identify, modulate, and express emotions [116]; moreover, a recent study found that individuals with high levels of EC are more prone to prefer immediate rewards [78]. Therefore, simply presenting contrary evidence to such individuals––who are prone to adopt a conspiracy mindset, risky behaviors, and refuse vaccination––will probably be unsuccessful in reversing their beliefs. Again, considering these aspects seems vital to developing effective strategies of intervention. 

The study partially confirmed our last hypothesis, showing that higher levels of personality pathology predicted the extraverted-maladaptive response pattern (D) to the pandemic emergency through the full mediation of low overall defensive functioning, poor certainty of others’ mental states, and high EC. Our findings suggest that individual maladaptive reactions to COVID-19, related restrictions, and vaccine hesitancy are influenced and characterized by identifiable psychological dimensions. Consistent with a psychodynamically oriented approach (cf. [26]), an accurate understanding of the interactions between personality characteristics and mental functioning abilities involved in poorly adaptive reactions to the pandemic has important implications for the development of effective mental health interventions [95]. 

This study presented some limitations that should be considered. First, the participant sample may not have been representative of the general population. Second, the survey usually used in the research field employed a snowball method via personnel contacts and social networks, which is not systematic and potentially vulnerable to biases. Third, the research design is cross-sectional; thus, it is not possible to determine causal relationships among all the psychological and behavioral dimensions included in the study. Despite these aspects, the results may significantly contribute to the implementation of interventions tailored to the specificities of individuals facing the pandemic. Notably, research supports the need to plan best practices to (1) increase the use of more mature coping strategies to deal with unacceptable or threatening feelings; (2) improve the ability to construct representations of one’s own and others’ internal mental states, as well as emotional regulation processes and relational interactions; and (3) promote the development of a more open and trusting stance for social learning in the context of meaningful and/or trusting relationships. 

## 5. Conclusions

The present empirical investigation revealed the presence of significant associations between response patterns to the pandemic containment measures and vaccination policies, and distinct aspects of personality and psychological functioning. Research suggests that identifying and understanding the complex individual mechanisms that hinder the ability to effectively cope with the difficulties associated with the emergency and adaptation to reality are essential to promote personalized psychological interventions. Undoubtedly, further studies in this research field are needed to better understand how distinct dimensions of psychological functioning influence individuals’ adaptation to rapid, unexpected, and adverse events such as those created by the pandemic.

The global emergency created by COVID-19 requires a cogent response from multidisciplinary mental health organizations. Recently (July 2022), Italy introduced a Free Bonus to facilitate access to psychological counseling for low-income populations. The findings of this study reinforce the need to strengthen mental health programs, incorporate ways of promoting mental well-being, identify forms of psychological distress that, if unheeded, could evolve into more serious and difficult-to-treat psychopathological conditions, and develop best practices for planning prevention activities and treatments focused on the specific qualities of individuals.

## Figures and Tables

**Figure 1 ijerph-19-14290-f001:**
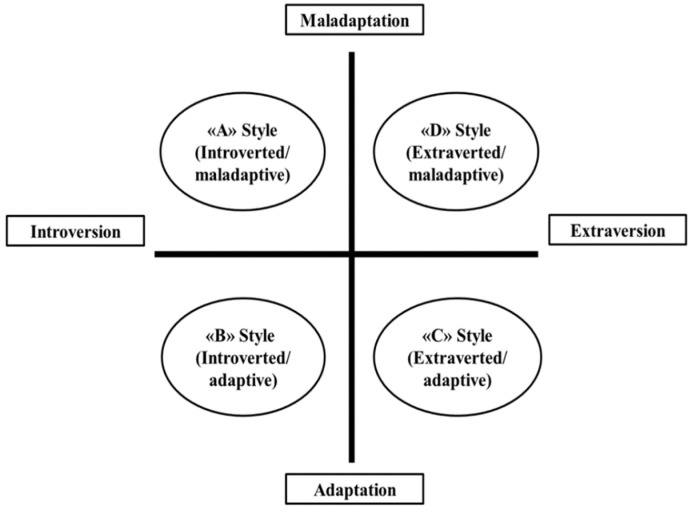
Figures Response Styles to the Pandemic and Lockdown(s) (RSPL).

**Figure 2 ijerph-19-14290-f002:**
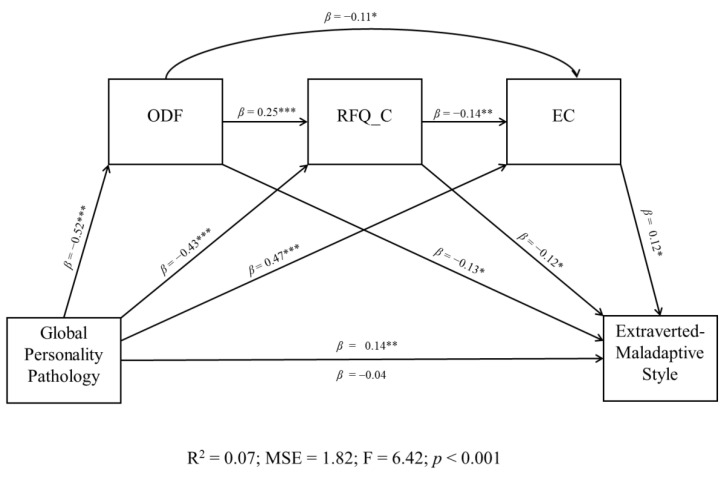
A Sequential Mediation Model Examining the Direct and Indirect Effects of Global Personality Pathology on Extraverted-Maladaptive Response Pattern to the COVID Pandemic and its Mitigation Strategies. Note. ODF = Overall Defensive Functioning; RFQ_C = Certainty About Mental States; EC = Epistemic Credulity. *β* = Standardized Beta Coefficient*;* R^2^ = R-squared; MSE = Mean Squared Error; F = F-distribution. * *p* ≤ 0.05. ** *p* ≤ 0.01. *** *p* ≤ 0.001.

**Table 1 ijerph-19-14290-t001:** Demographics and COVID-19-related characteristics.

Characteristics	N	%
Residence during the pandemic and lockdown(s)		
Urban setting	125	34.1
Suburban setting	132	36.0
Small town	82	22.3
Isolated dwelling	28	7.6
Significant job changes due to the pandemic		
Working remotely from home	187	50.9
Firing	5	1.4
Unemployment (already before the pandemic) with lower chances of finding a new job	34	9.3
Prolonged interruption in work activity	66	18.0
No change	69	18.8
Other	6	1.6
Economic difficulties due to the pandemic		
Yes	69	18.8
No	298	81.2
People living at home during lockdown(s)		
Close relatives	234	63.8
Friends	3	0.8
Partner	88	23.9
Alone	34	9.3
Other	8	2.2
Infection with COVID-19		
Yes	40	10.9
No	327	89.1
COVID-19 contagions among relatives and friends		
Yes	287	78.2
No	80	21.8
Bereavement among relatives and friends sick with COVID-19		
Yes	57	15.5
No	310	84.5

**Table 2 ijerph-19-14290-t002:** Bivariate Correlations Between Response Styles to the COVID-19 Pandemic and Lockdown(s) Assessed Using the RSPL ^a^ and the PID-5-BF ^b^ Personality Traits and Functioning (*N* = 367).

		RSPL			
Personality Traits and Functioning (PID-5-BF)		Introverted- Maladaptive Style (A)	Introverted- Adaptive Style (B)	Extraverted- Adaptive Style (C)	Extraverted- Maladaptive Style (D)
	*M (SD)*	2.94 (1.52)	4.15 (1.30)	3.96 (1.39)	1.99 (1.39)
Negative Affectivity	1.14 (0.59)	0.11 *	−0.06	0.03	0.09
Detachment	0.76 (0.65)	0.11 *	−0.07	−0.19 ***	−0.01
Antagonism	0.45 (0.43)	−0.01	0.03	0.01	0.16 **
Disinhibition	0.59 (0.51)	−0.02	−0.03	−0.02	0.17 ***
Psychoticism	0.56 (0.57)	0.06	0.05	−0.05	0.10
Global Personality Pathology	17.46 (9.99)	0.07	−0.03	−0.07	0.14 **

^a^ RSPL = Response Styles to the Pandemic and Lockdown(s). ^b^ PID-5-BF = Personality Inventory for DSM-5 Brief-Form. * *p* ≤ 0.05. ** *p* ≤ 0.01.*** *p* ≤ 0.001.

**Table 3 ijerph-19-14290-t003:** Bivariate Correlations Between Response Styles to the COVID-19 Pandemic and Lockdown(s) Assessed Using the RSPL ^a^ and the DMRS-SF-30 ^b^ Defense Mechanisms (*N* = 367).

		RSPL			
Defensive Functioning (DMRS-SF-30)		Introverted- Maladaptive Style (A)	Introverted- Adaptive Style (B)	Extraverted- Adaptive Style (C)	Extraverted- Maladaptive Style (D)
	*M (SD)*				
ODF	5.23 (0.54)	−0.08	0.07	0.05	−0.20 ***
Defense levels					
High-adaptive	45.21 (14.04)	−0.12 *	0.06	0.09	−0.20 ***
Obsessional	9.81 (6.48)	0.04	−0.02	−0.07	0.14 **
Neurotic	12.09 (5.00)	0.09	0.02	−0.06	−0.07
Minor-image distorting	8.75 (5.12)	0.04	−0.02	−0.05	0.12 *
Disavowal	10.10 (5.28)	0.17 ***	0.02	0.01	0.06
Major-image distorting	7.87 (4.63)	0.04	−0.09	−0.04	0.15 **
Action	6.17 (4.34)	−0.07	−0.08	−0.02	0.15 **
Defense mechanisms					
Suppression	5.54 (3.45)	−0.07	−0.06	−0.03	−0.17 ***
Sublimation	6.29 (4.10)	−0.07	0.12 *	−0.03	−0.03
Self-observation	7.14 (3.22)	−0.13 *	0.09	0.05	−0.14 **
Self-assertion	7.25 (3.97)	−0.13 *	0.09	0.04	−0.14 **
Humor	3.52 (3.06)	0.05	0.01	0.12 *	−0.15 **
Anticipation	5.89 (3.13)	−0.03	−0.06	0.07	−0.11 *
Altruism	6.23 (3.81)	−0.07	0.01	0.04	−0.08
Affiliation	3.34 (2.95)	−0.01	0.01	0.13 *	−0.02
Isolation of affects	3.26 (5.64)	0.03	−0.01	−0.06	0.11 *
Intellectualization	2.40 (2.29)	0.02	−0.01	−0.06	0.03
Undoing	4.15 (2.68)	0.02	−0.02	0.01	0.10
Repression	2.26 (2.48)	0.07	0.09	−0.01	−0.03
Dissociation	2.46 (1.89)	0.18 ***	−0.07	0.02	−0.02
Reaction Formation	3.87 (3.08)	−0.06	−0.04	−0.10	−0.07
Displacement	3.50 (2.59)	0.05	0.04	−0.02	−0.01
Omnipotence	2.83 (3.74)	−0.04	−0.04	−0.07	0.12 *
Idealization	2.96 (2.43)	−0.01	−0.01	−0.05	0.08
Devaluation	2.96 (2.66)	0.13 *	0.02	0.04	0.01
Denial	2.86 (2.64)	0.06	−0.01	0.04	0.04
Rationalization	2.80 (2.28)	0.07	0.06	0.03	0.07
Projection	2.03 (2.31)	0.11 *	0.01	−0.06	0.04
Autistic fantasy	2.40 (2.39)	0.15 **	0.01	0.01	−0.01
Splitting of self-image	2.48 (2.54)	0.02	−0.06	0.04	0.06
Splitting of others’ image	3.16 (2.57)	0.08	−0.02	−0.09	0.06
Projective identification	2.23 (2.34)	−0.02	−0.09	−0.02	0.17 ***
Passive aggression	1.15 (1.36)	−0.02	−0.06	−0.08	0.14 **
Help-rejecting complaining	2.94 (2.52)	−0.07	−0.07	−0.01	0.07
Acting out	2.08 (2.27)	−0.04	−0.04	0.02	0.12 *

^a^ RSPL = Response Styles to the Pandemic and Lockdown(s). ^b^ PID-5-BF = Personality Inventory for DSM-5 Brief-Form. Note. ODF = Overall Defensive Functioning. * *p* ≤ 0.05. ** *p* ≤ 0.01.*** *p* ≤ 0.001.

**Table 4 ijerph-19-14290-t004:** Bivariate Correlations Between Response Styles to the COVID-19 Pandemic and Lockdown(s) Assessed Using the RSPL ^a^, Mentalization Assessed Using the RFQ ^b^, and Epistemic Trust, Mistrust, and Credulity Assessed Using the ETMCQ ^c^ (N = 367).

		RSPL			
Mentalization and Epistemic Trust		Introverted- Maladaptive Style (A)	Introverted- Adaptive Style (B)	Extraverted- Adaptive Style (C)	Extraverted- Maladaptive Style (D)
	*M (SD)*				
RFQ					
RFQ_C	0.97 (0.61)	−0.16 **	0.03	0.03	−0.19 ***
RFQ_U	0.60 (0.65)	0.02	0.02	−0.01	0.03
ETMCQ					
ET	26.14 (4.68)	−0.01	0.06	0.13 *	−0.02
EM	18.64 (5.14)	0.11 *	−0.01	−0.01	0.08
EC	13.28 (5.90)	0.04	−0.01	−0.08	0.19 ***

^a^ RSPL = Response Styles to the Pandemic and Lockdown(s). ^b^ RFQ = Reflective Functioning Questionnaire. ^c^ ETMCQ = Epistemic Trust Mistrust Credulity Questionnaire. Note. RFQ_C = Certainty About Mental States; RFQ_U = Uncertainty About Mental States; ET = Epistemic Trust; EM = Epistemic Mistrust; EC = Epistemic Credulity. * *p* ≤ 0.05. ** *p* ≤ 0.01. *** *p* ≤ 0.001.

## Data Availability

Not applicable.

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
