# Peer review of "Personality, Defenses, Mentalization, and Epistemic Trust Related to Pandemic Containment Strategies and the COVID-19 Vaccine: A Sequential Mediation Model"

_ijerph, 2022, doi:10.3390/ijerph192114290_

Round 1
Reviewer 1 Report
It was a pleasure to have the opportunity to review this manuscript entitled "ersonality, Defenses, Mentalization, and Epistemic Trust Related to Pandemic Containment Strategies and the COVID-19 Vaccine: A Sequential Mediation Model". This study aimed to investigate the relationship between strategies adopted to manage the pandemic, vaccine hesitancy, and distinct features of personality and mental functioning.
I found the study very interesting for the psychological aspects investigated and well written. Both the introduction and conclusion sections are very comprehensive. In particular, the introduction allows readers to easily understand all the constructs involved in the study, while in the discussion each result obtained has been widely contextualized in the existing literature.
The manuscript, in my opinion, does not need much revision. Nevertheless, here are some tips:
- The term "mutual interaction" used in the description of mediation, could be misleading for the reader as it could refer to statistical bidirectionality not present in mediation, as well as "interaction" which could refer to the terms used in the moderation.
-Furthermore, it would be interesting to know if the authors found differences with respect to some socio-demographic variables, especially with respect to the origins as northern Italy was more affected than southern Italy.
Author Response
Reviewer: 1
Comments to the Author(s):
It was a pleasure to have the opportunity to review this manuscript entitled “Personality, Defenses, Mentalization, and Epistemic Trust Related to Pandemic Containment Strategies and the COVID-19 Vaccine: A Sequential Mediation Model”. This study aimed to investigate the relationship between strategies adopted to manage the pandemic, vaccine hesitancy, and distinct features of personality and mental functioning.
I found the study very interesting for the psychological aspects investigated and well written. Both the introduction and conclusion sections are very comprehensive. In particular, the introduction allows readers to easily understand all the constructs involved in the study, while in the discussion each result obtained has been widely contextualized in the existing literature.
We are very glad that you have appreciated our work. Thank you for your useful observations and comments that helped us improve our manuscript.
The manuscript, in my opinion, does not need much revision. Nevertheless, here are some tips:
- The term “mutual interaction” used in the description of mediation, could be misleading for the reader as it could refer to statistical bi-directionality not present in mediation, as well as "interaction" which could refer to the terms used in the moderation.
Sorry for the inaccuracy. We removed the expression “mutual interaction” from the manuscript.
- Furthermore, it would be interesting to know if the authors found differences with respect to some socio-demographic variables, especially with respect to the origins as northern Italy was more affected than southern Italy.
Following your helpful suggestion, we performed further statistical analyses to examine potentially significant differences in socio-demographic variables and reported all the findings in the “Results” section of the manuscript. Please see below:
“2.3. Statistical Analysis
Statistical analyses were performed using SPSS 26 for Windows (IBM, Armonk, NY). The relationships between response styles to pandemic containment measures (assessed using the dimensional scores obtained from RSPL vignettes) and socio-demographic variables (e.g., gender, age, geographic provenance) were analyzed using Multivariate Analysis of Variance (MANOVA) and bivariate correlations (r, two-tailed).”
[omissis]
“Analyses that examined the relationships between response styles to pandemic containment measures (assessed using RSPL dimensional scores) and specific socio-demographic characteristics (including gender, geographic provenance, and socio-economic level) did not find significant results. MANOVA showed no effect of gender on individuals’ reactions to COVID-19 emergency, Wilks’s l = .99, F(4, 362) = 1.189, p = .32, h2 = .013, Cohen’s d = 0.23. More in detail, female participants (M introverted/maladaptive style = 2.92, SD = 1.50; M introverted/adaptive style = 4.08, SD = 1.34; M extraverted/adaptive style = 4.04, SD = 1.37; M extraverted/maladaptive style = 2.05, SD = 1.43) did not report significantly different response patterns than male participants (M introverted/maladaptive style = 3.00, SD = 1.57; M introverted/adaptive style = 4.28, SD = 1.20; M extraverted/adaptive style = 3.79, SD = 1.42; M extraverted/maladaptive style = 1.86, SD = 1.30).
Moreover, analysis showed no impact of geographic provenance on response styles to the pandemic, Wilks’s l = .97, F(8, 722) = 1.390, p = .19, h2 = .015, Cohen’s d = 0.25. Participants from northern Italy (M introverted/maladaptive style = 2.83, SD = 1.52; M introverted/adaptive style = 4.19, SD = 1.26; M extraverted/adaptive style = 3.88, SD = 1.40; M extraverted/maladaptive style = 2.07, SD = 1.49) did not report significantly different responses than those from central Italy (M introverted/maladaptive style = 3.08, SD = 1.54; M introverted/adaptive style = 4.11, SD = 1.35; M extraverted/adaptive style = 4.14, SD = 1.34; M extraverted/maladaptive style = 1.90, SD = 1.22) or from southern Italy and the islands (M introverted/maladaptive style = 3.12, SD = 1.42; M introverted/adaptive style = 4.00, SD = 1.39; M extraverted/adaptive style = 3.65, SD = 1.47; M extraverted/maladaptive style = 1.85, SD = 1.35).
No effect of socio-economic level was also revealed, Wilks’s l = .97, F(8, 722) = 1.385, p = .20, h2 = .015, Cohen’s d = 0.25. Participants from the upper class (M introverted/maladaptive style = 2.88, SD = 1.39; M introverted/adaptive style = 4.38, SD = 1.16; M extraverted/adaptive style = 4.18, SD = 1.27; M extraverted/maladaptive style = 1.56, SD = 1.02) did not report significantly different responses than those from middle (M introverted/maladaptive style = 2.91, SD = 1.51; M introverted/adaptive style = 4.13, SD = 1.31; M extraverted/adaptive style = 3.95, SD = 1.39; M extraverted/maladaptive style = 2.01, SD = 1.40) and lower classes (M introverted/maladaptive style = 3.24, SD = 1.72; M introverted/adaptive style = 4.05, SD = 1.32; M extraverted/adaptive style = 3.86, SD = 1.51; M extraverted/maladaptive style = 2.21, SD = 1.54). Finally, no significant relationship was found between age and pandemic emergency responses, except for the negative correlation with extrovert-adaptive style (r = -.24, p < .001).”
Reviewer 2 Report
The topic is important and I appreciate the authors' efforts. I encourage continued research and submission of a future manuscript. However, I am sorry to conclude that the potential contributions of the present manuscript are unclear. There is no clear theoretical or practical rationale for the investigation - how, exactly, does it aim to further the literature and extend it conceptually, and what, exactly are the practical ramifications?'
First, it draws on so many different constructs and theoretical frameworks that it cannot dig deep enough to make a signficant contribution to either of them, and a far more comprehensive conceptual underpinning would be needed to synthesize them. Second, it does not clearly discriminate between pandemic responses and vaccine responses, which not only conceptually, but also pragmatically, seem like to very different things. Hence, the conclusions come across a bit muddled.
Also, it relies on a rather small online sample, which would partly explain why more complex analyses are not feasible (for lack of power), but also limit the empirical contribution.
I do think the topic is highly important, and the authors prove to be knowledgeable in both the theory and the method, so I am certain a new investigation which narrows the scope and expands the conceptual elaboration, will make for very interesting future contributions. Thank you for an interesting read and best of luck with the continued work!
Author Response
Reviewer: 2
Comments to the Author(s):
The topic is important and I appreciate the authors’ efforts. I encourage continued research and submission of a future manuscript. However, I am sorry to conclude that the potential contributions of the present manuscript are unclear. There is no clear theoretical or practical rationale for the investigation - how, exactly, does it aim to further the literature and extend it conceptually, and what, exactly are the practical ramifications?
First, it draws on so many different constructs and theoretical frameworks that it cannot dig deep enough to make a significant contribution to either of them, and a far more comprehensive conceptual underpinning would be needed to synthesize them. Second, it does not clearly discriminate between pandemic responses and vaccine responses, which not only conceptually, but also pragmatically, seem like to very different things. Hence, the conclusions come across a bit muddled.
Also, it relies on a rather small online sample, which would partly explain why more complex analyses are not feasible (for lack of power), but also limit the empirical contribution.
I do think the topic is highly important, and the authors prove to be knowledgeable in both the theory and the method, so I am certain a new investigation which narrows the scope and expands the conceptual elaboration, will make for very interesting future contributions. Thank you for an interesting read and best of luck with the continued work!
Before discussing your comments and observations, which helped us to improve the manuscript, we would like to better clarify the rationale of our research project.
This is the first study that sought to explore the role of different psychological dimensions (closely related to traumatic events) in coping with the pandemic emergency through the lens of a psychodynamically-oriented theoretical model (e.g., Bateman & Fonagy, 2012; Gabbard, 2014; Kernberg, 1984; Lingiardi & McWilliams, 2017).
This empirical investigation aimed: (1) to fill a gap in the international empirical literature and seek to expand the scientific knowledge on “Mental Health in the Time of COVID-19” (in line with the purpose of this Special Issue), especially taking into account that most studies have focused on specific symptom patterns or clinical conditions, while only a more limited amount of research has focused on distinct aspects of intra- and interpersonal functioning associated with pandemic emergency and vaccine hesitancy; (2) to provide a sufficiently articulated representation of how personality, defense mechanisms, mentalization, and epistemic trust (two closely interrelated constructs in light of theoretical and empirical contributions; e.g., Campbell et al. , 2020; Fonagy et al., 2019; Luyten et al., 2020) can influence individuals’ adaptation to unexpected, challenging, and dramatic situations; and, consequently, (3) to promote personalized interventions that are tailored on the unique and distinctive characteristics of individuals, helping them respond to the deep-seated vulnerabilities that the pandemic has brought to light.
Notably, according to your useful comments, in the “Introduction” section of the manuscript we have better clarified the rationale behind the research design and pointed out that its theoretical framework is based on a psychodynamically-oriented model. Secondly, in the “Method” section we have better specified that response patterns to the pandemic and attitudes toward COVID-19 vaccines are different constructs. As such, they have been investigated in different ways.
Finally, in line with your persuasion, we are continuing to invest in this line of research. We have planned a second study with a larger sample to corroborate and generalize the promising results of the present investigation. Again, we thank you for your encouragement and support in the effort to foster more complex empirical investigations in the field.
Reviewer 3 Report
I reviewed the manuscript entitled "Personality, Defenses, Mentalization, and Epistemic Trust Related to Pandemic Containment Strategies and the COVID-19 Vaccine: A Sequential Mediation Model". The paper is well-written and interesting for the readers, the tables and figures were clear. The subject of the study is up to date, because the knowledge regarding psychological characteristics and covid-19 is still evolving. Moreover, the results of the project could be used in clinical practice. In my opinion, the manuscript is of a high scientific quality, however, some considerations need to be made.
The manuscript would benefit from a mitigation of some assumptions such as:
1. page 2 line 58 "anti-vaccine activist..." (not all the activist can be described by this sentence)
2. The first part of the conclusion page 15 line 625 "the present empirical investigation..)
Both this statements should be toned-down.
Author Response
Reviewer: 3
Comments to the Author(s):
I reviewed the manuscript entitled "Personality, Defenses, Mentalization, and Epistemic Trust Related to Pandemic Containment Strategies and the COVID-19 Vaccine: A Sequential Mediation Model". The paper is well-written and interesting for the readers, the tables and figures were clear. The subject of the study is up to date, because the knowledge regarding psychological characteristics and covid-19 is still evolving. Moreover, the results of the project could be used in clinical practice. In my opinion, the manuscript is of a high scientific quality, however, some considerations need to be made.
We are really pleased with your appreciation of our work.
The manuscript would benefit from a mitigation of some assumptions such as:
- page 2 line 58 "anti-vaccine activist..." (not all the activist can be described by this sentence)
- The first part of the conclusion page 15 line 625 "the present empirical investigation..)
Both this statements should be toned-down.
Thank you for your thoughtful observation. As suggested, we have reformulated and “mitigated” specific assumptions in the “Introduction” and “Discussion” sections of the manuscript.
Please, see below:
“The Italian governmental bodies and other organizations have tried to adopt many strategies to incentivize vaccination; however, protests, demonstrations, and strikes have rapidly spread—similarly to what happened in other countries. Some anti-vaccine activists, seemingly influenced by a conspiracy mindset, have also developed unfounded beliefs about COVID-19 vaccines based on misunderstood or misrepresented (un)scientific or religious theories.”
In the “Conclusion” we have also remarked the need for further empirical studies in this research field.
“The present empirical investigation revealed the presence of significant associations between response patterns to the pandemic containment measures, vaccination policies, and distinct aspects of personality and psychological functioning. [omissis] Undoubtedly, further studies in this research field are needed to better understand how distinct dimensions of psychological functioning influence individuals' adaptation to rapid, unexpected, and adverse events such as those created by the pandemic.
Round 2
Reviewer 1 Report
I appreciate that the authors have addressed all the recommendations. The manuscript is now suitable for the publication.
Reviewer 3 Report
I welcome the revisions you made in your manuscript, and I appreciate the consideration you gave to my comments